# Potentially unsafe doses of local anesthetics in axillary brachial plexus block: A single-center retrospective cohort study

Mélanie Suppan[1,2]*, Caroline Flora Samer[2,3], Georges Louis Savoldelli[1,2]

1 Division of Anaesthesiology, Department of Acute Care Medicine, Geneva University Hospitals, Geneva, Switzerland, 2 Department of Anaesthesiology, Pharmacology, Intensive Care and Emergency Medicine, Faculty of Medicine, University of Geneva, Geneva, Switzerland, 3 Division of Clinical Pharmacology and Toxicology, Department of Acute Care Medicine, Geneva University Hospitals, Geneva, Switzerland

* melanie.suppan@hug.ch

## Abstract

Local anesthetic systemic toxicity is a rare but potentially life-threatening complication of regional anesthesia that can occur when high doses of local anesthetics are administered. This study aimed to evaluate the frequency of local anesthetic doses exceeding safe thresholds in axillary brachial plexus blocks using four different calculation methods. This retrospective study analyzed 2395 patients who underwent axillary brachial plexus block between 2017 and 2021 at Geneva University Hospitals. Four progressively more conservative sets of dosing rules were systematically applied. These included standard package insert recommendations, weight-based limits using actual weight, weight-based limits using ideal body weight, and consensus-based rules adapted to patients' comorbidities and treatments. For local anesthetic mixtures, proportional calculations were applied to determine cumulative maximum safe doses. Using the most conservative calculation method, local anesthetic doses exceeded maximum safe thresholds in 64.8% of cases, compared to 29.5% using package insert recommendations. Potentially unsafe doses were consistently more frequent with local anesthetic mixtures (85.4%) compared to single agents (32.4%) across all calculation methods. Symptoms compatible with local anesthetic systemic toxicity occurred in 19 patients (0.79%), with severe manifestations in 9 cases (0.38%). No significant relationship was found between these symptoms and potentially unsafe doses, regardless of the calculation method used. This study reveals substantial variation in local anesthetic dosing practices for axillary brachial plexus blocks. Rates of potentially unsafe doses varied significantly depending on the criteria applied. The consistent pattern of higher rates of potentially unsafe doses with mixture use highlights opportunities for practice standardization and improved safety protocols in regional anesthesia. The multiple calculation approaches allow clinicians to compare findings with their own institutional practices.

**Data availability statement:** The anonymized dataset containing all data of the patients included in the analysis is available in the Yareta repository (University of Geneva) at: https://doi.org/10.26037/YARETA:3HCQXRX5ZFGULNWRCIBUCJGZIU.

**Funding:** The author(s) received no specific funding for this work.

**Competing interests:** All authors were involved in the development of the LoAD Calc application, which was developed for research purposes only. This application is not monetized and will not be monetized in the future. The calculation rules from this application were used to define the most conservative calculations described in this manuscript. This does not alter our adherence to PLOS ONE policies on sharing data and materials.

## Introduction

Regional anesthesia, particularly peripheral nerve blocks, has become integral to modern anesthetic practice, offering effective pain control while reducing opioid requirements [1]. Axillary brachial plexus block is one of the most commonly performed procedures, valued for its reliability and ease of learning [2]. This technique involves injecting local anesthetics (LA) near the brachial plexus to provide anesthesia for upper limb surgery.

Local anesthetic systemic toxicity (LAST) is a rare but potentially life-threatening complication that can occur when LA enter systemic circulation in sufficient quantities. LAST manifests through symptoms ranging from mild neurological signs such as perioral numbness to severe complications including seizures and cardiovascular collapse [1]. While ultrasound guidance has improved safety by reducing vascular puncture rates, LAST continues to occur with reported incidence rates varying from 0.04% to nearly 1% depending on the clinical setting and diagnostic criteria [1,3–5].

The prevention of LAST relies heavily on appropriate LA dosing, yet establishing safe dose thresholds presents significant challenges. Maximum recommended doses are expressed through various approaches including weight-based limits (mg/kg), absolute maximum values (mg), or combinations of both [6]. These definitions vary substantially between countries and clinical settings, creating inconsistencies that impede standardized dosing practices [7–9]. Furthermore, patient-specific factors such as age, renal and hepatic function, and concurrent medications can alter LA metabolism, complicating dose determination [10].

Several key controversies persist regarding safe LA dosing. First, there is ongoing debate about whether actual body weight, ideal body weight, lean body weight or alternative calculations should be used, particularly in obese patients [11,12]. Second, calculating maximum safe doses for LA mixtures remains unclear, with limited guidance on accounting for potentially additive toxic effects [13]. Third, the extent to which patient-specific factors should modify standard recommendations lacks consensus [9,14].

Axillary brachial plexus blocks present particular challenges due to anatomical factors. Historically, this technique required high volumes to ensure adequate spread around multiple nerve branches [15]. Although ultrasound guidance enables more targeted delivery [16], many practitioners continue using substantial volumes to ensure success. The proximity to major vessels, combined with the need for multiple injections, increases risks of intravascular injection and systemic absorption [2]. Furthermore, the practice of mixing LA to combine the rapid onset of one agent with the prolonged duration of another, while common in some institutions, remains variable and debated, with limited evidence supporting superior clinical outcomes compared to single agents [17]. When used, such mixtures further complicate dose calculations [18].

Given these challenges, comprehensive data on current LA dosing practices are needed. To our knowledge, no previous study has systematically evaluated LA dosing using multiple calculation methods that reflect diverse clinical approaches. The primary aim of this study was to evaluate the frequency of LA doses exceeding

calculated safe thresholds in axillary brachial plexus blocks, using four different calculation methodologies. Secondary aims were to identify factors associated with potentially unsafe dosing and assess relationships between calculated dose exceedances and clinical manifestations of LAST.

The hypothesis was that a significant proportion of patients would receive potentially unsafe doses according to various calculation methods, with proportions varying substantially depending on the calculation approach used, thereby demonstrating the need for standardized dosing protocols in regional anesthesia practice.

## Materials and methods

### Study design

This was a retrospective cohort study carried out and reported according to the Strengthening the Reporting of Observational Studies in Epidemiology (STROBE) statement [19]. Approval by the regional ethics committee (Commission Cantonale d'Ethique de la recherche CCER – Req 2022−01195, Geneva, Switzerland, Chairperson Prof B. Hirschel) was obtained on 16/08/2022. Only adult patients (aged 18 or older) were included in this study. A written general consent form for data reuse has been in place at our institution since 2017. However, given the retrospective nature of this study using anonymized medical records that posed no potential harm to participants, the ethics committee waived the requirement for individual informed consent and authorized data reuse even without a signed general consent form.

For sample size calculation, to avoid fragility, precision was increased by dividing the estimated prevalence (0.01) by 5. It was thus estimated that, to detect a 1% dose difference with high precision (0.002) and 95% power, a sample of 2234 axillary brachial plexus blocks procedures would be necessary [20].

### Definition of maximum safe doses

Four different calculation methods were selected to represent a spectrum of approaches encountered in clinical practice, from the simplest to the most individualized. These range from basic package insert recommendations, commonly used as a default reference, to actual weight-based calculations reflecting routine clinical practice, to ideal body weight-based calculations addressing pharmacokinetic concerns in overweight and obese patients, and finally to conservative consensus-based rules incorporating patient-specific factors known to affect LA metabolism (age, organ dysfunction, drug interactions) that are rarely systematically considered in everyday practice. Comparing these progressively more conservative approaches allows clinicians to benchmark their own institutional practices against multiple standards and illustrates how the assessment of dosing safety varies depending on the criteria applied.

The first method used standard LA package inserts recommendations, regardless of individual patient characteristics: 150 mg for levobupivacaine, 250 mg for ropivacaine and 300 mg for lidocaine. We then applied the most conservative weight-based doses found in the literature for our three subsequent analyses. Specifically, maximum LA dose limit of 3 mg/kg was used for lidocaine and ropivacaine while a limit of 2 mg/kg was used for levobupivacaine [11,21,22]. If epinephrine was added, these limits were increased to 7 mg/kg for lidocaine and to 3 mg/kg for levobupivacaine, while the limit for ropivacaine remained unchanged [23]. Other additives were not taken into account since data were too scarce to determine their effect, if any, on maximum safe LA doses [24].

The second analysis was carried out taking only the patient's actual weight (AW) into account. For the third, ideal body weight (IBW) was used instead of the patient's actual weight and was computed using Devine's formula [12]. This formula was chosen due to its widespread use in clinical pharmacology and its standard implementation in our institution's anesthesia records, where it is automatically computed for each patient.

For the last, most conservative set of calculation rules, relevant patient characteristics and concurrent treatments were included [14]. The comprehensive rules used to carry out these calculations have already been described [24]. Briefly, the first step was to determine the calculation weight (CW). To this end, the body mass index (BMI) and ideal body weight (IBW) according to Devine's formula were determined for each patient. Calculation weights were limited to a maximum of

70 kg, even for patients weighing more than 70 kg, to establish a ceiling effect preventing excessive absolute LA doses while maintaining appropriate weight-based dosing for patients of lower weight [11]. This was achieved through the following sequence:

AW ≤ 70 kg, BMI < 30 kg/m2 and IBW> AW → CW = AW,

AW ≤ 70 kg, BMI < 30 kg/m2 and IBW ≤ AW → CW = IBW,

AW ≤ 70 kg and BMI ≥ 30 kg/m2 → CW = IBW,

AW > 70 kg and IBW > 70 kg → CW = 70 kg,

AW > 70 kg and IBW ≤ 70 kg → CW = IBW.

Patient characteristics altering LA metabolism were then considered to determine whether these limits should be lowered further: age 70 or higher [9], renal dysfunction (glomerular filtration rate < 50 ml/min) [25], hepatic dysfunction (prothrombin time < 50%), heart failure (left ventricular ejection fraction ≤ 30%), pregnancy, and concurrent intake of medications altering LA metabolism (major CYP1A2 or CYP3A inhibitors, such as ciprofloxacin or macrolides, respectively) [9,14]. Since there are no clear, evidence-based calculation rules to take the impact of these characteristics into account, conservative, consensus-based rules were established by a panel including expert anesthesiologists and clinical pharmacologists. If one of the aforementioned characteristics was present, the maximum safe dose was reduced by 20%; if two or more were present, it was reduced by 30% [24].

### Inclusion and exclusion criteria

The computerized files of all patients aged 18 or older who had undergone axillary brachial plexus block between 01/01/2017 and 31/12/2021 at Geneva University Hospitals (HUG) were included in this study. Files belonging to patients who had refused the reuse of their medical data and incomplete files, i.e., those which did not contain enough data to calculate the primary outcome (weight, height, details of LA use), were excluded. Files were also excluded when LA other than lidocaine, levobupivacaine or ropivacaine were administered, when unusual concentrations were used (i.e., concentrations other than 0.25%, 0.375%, or 0.5% for levobupivacaine and ropivacaine, and other than 0.5% or 1% for lidocaine), or if more than 2 LA had been given. Patients whose height was lower than 153 cm, thus preventing IBW calculation according to Devine's formula, were also excluded [12]. Additionally, patients were excluded if they underwent general anesthesia, if only a single nerve belonging to the axillary brachial plexus was blocked, or if other types of regional anesthesia were used.

### Outcomes

The primary outcome was the proportion of patients who received LA doses higher than the maximum safe dose according to most conservative, full set of calculation rules described above.

Secondary outcomes were the proportion of patients who received LA doses higher than calculated according to package inserts' recommendations, to AW and to IBW; the mean difference, in mg, between the doses administered and maximum safe doses according to package inserts' recommendations, AW, IBW and the full calculation procedure; the incidence of symptoms compatible with LAST; the incidence of vascular puncture; and the rate of neurostimulation and ultrasound use.

The influence of patient age and sex, and of operator sex and experience, were also analyzed.

### Data collection and availability

Data were accessed for research purposes between 20/08/2022 and 15/12/2023, following ethics committee approval obtained on 16/08/2022. An electronic case report form (eCRF) was created using a REDCap electronic data capture tool hosted at HUG. All automatically extracted data were then imported, and all files were then manually reviewed to insert relevant data which had not or could not be automatically extracted. This included a review of intraoperative and follow-up

notes to identify the presence of potential LAST symptoms. We classified symptoms based on established criteria in the literature, which categorize manifestations as mild-to-moderate (perioral numbness, metallic taste, confusion, muscle twitching, etc.) or severe (seizures, loss of consciousness, respiratory depression, cardiac arrhythmias, severe hypotension, and cardiac arrest) [1].

During data collection, authors had access to information that could potentially identify individual participants. However, all data were subsequently anonymized and assigned coded identifiers before analysis to ensure patient confidentiality. The anonymized dataset containing all data of the patients included in the analysis has been uploaded to the Yareta repository [26]. This final dataset contains no personally identifiable information, ensuring complete patient privacy protection.

## Statistical analysis

A comma separated values (CSV) file was exported from REDCap and imported in Stata 17 (StataCorp LLC, College Station, Texas, USA), which was used for data curation and statistical analysis.

Descriptive statistics (n, %) were used to detail the basic demographic and clinical data, including the doses of LA administered. Given the sample size, these data were reported as mean±SD.

Since systemic toxicity of LA is considered additive when multiple agents are used [13], proportional dose adjustments were applied for mixtures. The maximum safe dose of the first LA was calculated. If this dose already exceeded safety thresholds, it was considered potentially unsafe regardless of additional agents. If the first LA dose was within safe limits, the percentage of the maximum safe dose already utilized was determined. The remaining "safety margin" was then applied proportionally to the second agent. For example, if a patient received 70% of their calculated maximum safe dose of ropivacaine, only 30% of the maximum safe dose of lidocaine could be administered without potentially exceeding cumulative toxicity thresholds.

For weight-based analyses, the maximum safe dose of the first LA used was determined based on the patient's AW, IBW, and finally according to the most conservative set of calculation rules described above. This maximum safe dose was considered equal to 100% of each of these calculated doses. The proportion of patients who had received a potentially unsafe dose (i.e., a dose over 100% of the calculated dose) and the related 95% CI were then reported accordingly. Additionally, when subgroups contained fewer than 5 patients, doses exceeding the maximum safe dose (calculated according to standard guidelines) were reported as median (Q1:Q3) in mg rather than mean values.

The Chi-squared test was used to search for a difference in potentially unsafe dosing between patients who had received a single LA and those who had received a mixture.

The influence of patient sex, patient age, operator experience and operator sex were assessed through multivariable logistic regression [27]. The effect size of each of these variables was reported through 95% CIs. The variables were selected according to their clinical relevance and to data availability. Weight, comorbidities, and interacting drugs were not included as independent predictors because they are already incorporated into the outcome calculation, which would introduce collinearity. There was no risk of overfitting given the high proportion of potentially unsafe doses. Multicollinearity was ruled out using Spearman's test, and log-linearity was checked graphically. Goodness-of-fit was assessed using the Akaike Information Criterion (AIC). The AIC was used because, since the weight-based calculation approaches used different outcome definitions rather than different predictor sets, they are non-nested and cannot be compared using likelihood ratio tests.

No imputation methods were used for missing data. Analyses were performed only on available data (complete case analysis), and the extent of missing data for each variable is reported. P values < 0.05 were considered significant.

## Results

A total of 3767 files were identified. After excluding 1372 files, 2395 (63.6%) were finally analyzed (Fig 1). Patient and operator characteristics are detailed in Table 1. Of the 2395 patients analyzed, 930 (38.8%) received a single LA agent

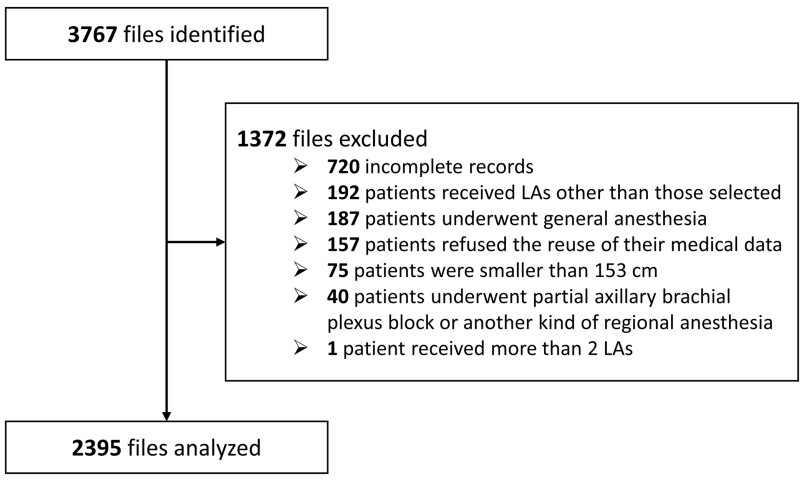

**Fig 1. Study flowchart.**

**Table 1. Patient characteristics (stratified by patient sex) and operator characteristics (stratified by operator sex).**

| Patient characteristics | | | |
|---|---|---|---|
| Sex (n, %) | Overall (n = 2395) | Female: (n = 982; 41.0%) | Male: (n = 1413; 59.0%) |
| Age (years ± SD) | 49 ± 19 | 57 ± 19 | 44 ± 17 |
| Height (cm ± SD) | 171 ± 9 | 164 ± 6 | 177 ± 7 |
| Weight (kg ± SD) | 73.9 ± 15.6 | 65.8 ± 13.9 | 79.5 ± 14.2 |
| Body mass index (kg/m$^2$ ± SD) | 25.1 ± 4.6 | 24.6 ± 5.1 | 25.4 ± 4.2 |
| Ideal body weight (kg ± SD) | 65.4 ± 10.2 | 55.6 ± 5.4 | 72.1 ± 6.7 |
| Relevant comorbidities (n, %) | | | |
| Renal failure (GFR[1] < 50 ml/min) | 101 (4.2%) | 41 (4.2%) | 60 (4.3%) |
| Liver failure (PT[2] < 50%) | 0 (0%) | 0 (0%) | 0 (0%) |
| Congestive heart failure (LVEF[3] ≤ 30%) | 6 (0.3%) | 3 (0.3%) | 3 (0.2%) |
| Pregnancy (n, %) | 6 (0.3%) | 6 (0.6%) | 0 (0.0%) |
| Treatment altering LA[4] metabolism at the time of the intervention[5] (n, %) | 174 (7.3%) | 78 (7.9%) | 96 (6.8%) |
| **Operator characteristics** | | | |
| Sex (n, %) | Overall (n = 2395) | Female: (n = 1036; 43.3%) | Male: (n = 1359;56.7%) |
| Experience (n, %) | | | |
| Resident | 1347 (56.2%) | 594 (57.3%) | 753 (55.4%) |
| Registrar | 623 (26.0%) | 368 (35.5%) | 255 (18.8%) |
| Consultant | 394 (16.5%) | 74 (7.1%) | 320 (23.6%) |
| Rotation from another specialty | 31 (1.3%) | 0 (0.0%) | 31 (2.3%) |

[1]GFR: Glomerular Filtration Rate; [2]PT: Prothrombin Time, [3]LVEF: Left Ventricular Ejection Fraction; [4]LA: local anesthetic; [5]Major CYP1A2 inhibitors such as ciprofloxacin, norfloxacin, and fluvoxamine, and major CYP3A inhibitors, such as azole antifungals, macrolides, calcium channel blockers, HIV antiretroviral therapy, and tyrosine kinase inhibitors. Totals may not equal 100% due to rounding.

and 1465 (61.2%) received a mixture of two agents. Among single-agent blocks, ropivacaine was used in 873 patients (93.9%), levobupivacaine in 46 patients (4.9%), and lidocaine in 11 patients (1.2%). The vast majority of mixtures consisted of ropivacaine with lidocaine (1430/1465, 97.6%), followed by levobupivacaine with lidocaine (31/1465, 2.1%), and other combinations (4/1465, 0.3%) Epinephrine was not used as an adjuvant in any of the 2395 blocks analyzed.

The mean dose of ropivacaine administered as a single agent (n=873) was 158±35 mg (2.2±0.6 mg/kg AW; 2.5±0.7 mg/kg IBW). For levobupivacaine (n=46), doses of 127±30 mg (1.7±0.5 mg/kg AW; 2.0±0.6 mg/kg IBW) were given, while doses of 302±37 mg (4.0±0.8 mg/kg AW; 4.7±1.0 mg/kg IBW) were administered when lidocaine was used alone (n=11).

Most mixtures consisted in ropivacaine or levobupivacaine with lidocaine (1430/1465, 97.6%, and 31/1465, 2.1% respectively). In the first case, 92±22 mg of ropivacaine (1.3±0.4 mg/kg AW; 1.4±0.4 mg/kg IBW) with 171±42 mg of lidocaine (2.4±0.8 mg/kg AW; 2.7±0.8 mg/kg IBW) were administered. In the second case, 99±25 mg of levobupivacaine (1.4±0.4 mg/kg AW; 1.5±0.4 mg/kg IBW) with 154±60 mg of lidocaine (2.2±0.9 mg/kg AW; 2.4±1.0 mg/kg IBW) were given.

According to package inserts' recommendations, the proportion of patients who received a potentially unsafe dose was 29.5% (706/2395). The probability of receiving a potentially unsafe dose was higher when a mixture was used than when a single agent was administered (697/1465, 47.5% vs 9/930, 1.0%, P<0.001).

Using weight-based analyses, the proportion of potentially unsafe LA doses were of 49.6% (1187/2395) and 58.4% (1398/2395) when AW and IBW were taken into account, respectively. Once again, the use of mixtures was associated with a higher probability of receiving a potentially unsafe dose (P<0.001 in both cases).

With the full, most conservative set of calculation rules, the proportion of patients who received a potentially unsafe dose was 64.8% (1552/2395). Among the 930 patients who received a single LA, 301 (32.4%) were given potentially unsafe doses (Fig 2). In these cases, ropivacaine (N=272) exceeded safe doses by a median of 22.4 (10.0: 40.0) mg, levobupivacaine (N=25) by a median of 30.6 (18.8: 56.0) mg, and lidocaine (N=4) by a median of 87.3 (62.9: 131.8) mg. Most patients who received LA mixtures were given potentially unsafe doses (1251/1465, 85.4%). The probability of receiving a potentially unsafe dose was significantly higher when a mixture was used than when a single agent was administered (85.4% vs. 32.4%, P<0.001). The dose of the first LA given already exceeded recommendations even before the second LA was added in 37 cases (ropivacaine in 25 cases and levobupivacaine in 12). Of the 1462 patients

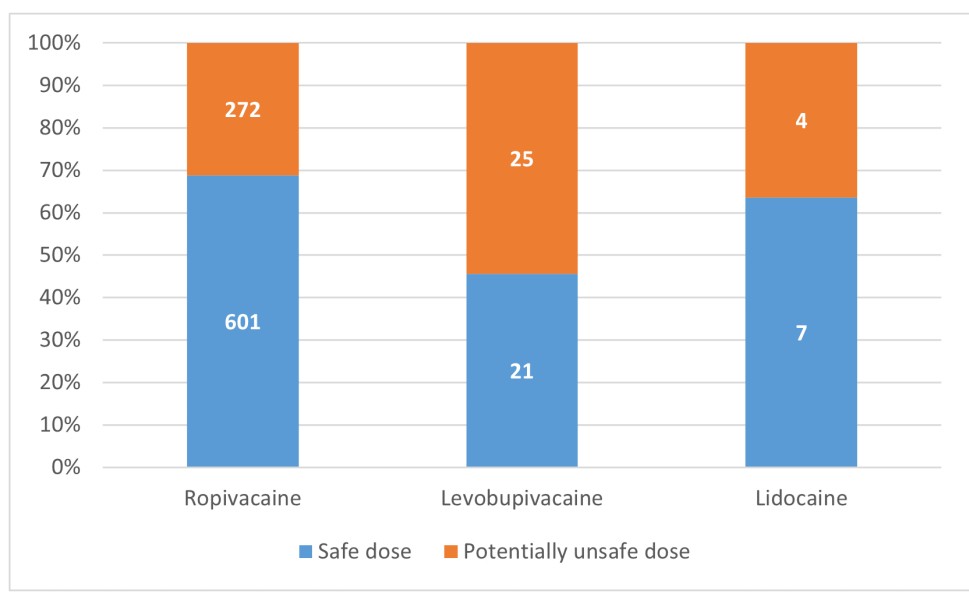

**Fig 2. Proportion of patients receiving potentially unsafe local anesthetic doses when a single agent was used, according to the most conservative calculation rules.**

who received lidocaine as second agent, 1250 received potentially unsafe doses (1250/1462, 85.5%). The mean excess lidocaine dose above the calculated maximum dose using the full calculation rules was 85.1 ± 48.8 mg.

The probability of potentially unsafe LA dose was not influenced by operator experience (Table 2). When the full calculation rules were considered, female patients and older patients were more likely to receive potentially unsafe LA dose, and female anesthesiologists were more likely to administer a potentially unsafe dose.

The association of patient and operator sex with the probability of potentially unsafe dose were consistent and of similar magnitude when IBW or AW were considered. The association with older age was still present when IBW was considered but disappeared when AW was used (Table 2).

Symptoms compatible with LAST were found in 19 patients (0.79%, 19/2395), and were considered severe in nine cases (0.38%, 9/2395). Cardiovascular symptoms were present in 13 patients (nine of which had severe symptoms) while neurological symptoms were described in nine patients (with only one case deemed severe). There were 13 cases of LAST-compatible symptoms in patients with potentially unsafe LA doses and six in patients who received adequate LA doses. There was no statistically significant association between the occurrence of LAST symptoms and potentially unsafe LA dose when considering either IBW, AW or full rules (P=0.373, P=0.505 and P=0.742, respectively). There were no fatalities.

Vascular puncture was reported in 28 of 2387 analyzable cases (1.17%). Eight records did not contain enough information to be included in this analysis. None of the patients in whom vascular puncture was reported presented LAST-compatible symptoms.

Neurostimulators were used in 1618 of 2102 analyzable cases (77.0%), while an ultrasound was used in 2086 of 2097 cases (99.5%). These denominators reflect the fact that some records did not contain enough data to ascertain the use of neurostimulators or ultrasound devices (293 and 298, respectively).

## Discussion

This retrospective cohort study demonstrates substantial variation in LA dosing practices for axillary brachial plexus blocks, with the proportion of potentially unsafe doses ranging from 29.5% to 64.8% depending on the calculation criteria applied. These findings primarily highlight the complexity anesthesiologists face when determining safe LA doses and the need for greater standardization in dosing practices. The use of four different calculation methods provides comprehensive insight into current dosing practices and allows clinicians to compare these findings with their own institutional approaches.

Despite the use of ultrasonography, which reduces vascular puncture, enhances nerve block success, and allows effective blockade to be achieved with lower LA volumes, a significant portion of potentially unsafe doses were observed across all methods [28]. Effective blockade can be achieved with lower LA volumes [16,29], but the comparatively lower success rate of the axillary brachial plexus block may lead anesthesiologists to administer higher LA volumes, inadvertently increasing the risk of LAST [2,30,31].

**Table 2. Predictors of potentially unsafe local anesthetic dosing: multivariable logistic regression analysis.**

| | Full rules (AIC: 2944) | | IBW (AIC: 3123) | | AW (AIC: 3258) | |
|---|---|---|---|---|---|---|
| | OR (95%CI) | P-value | OR (95%CI) | P-value | OR (95%CI) | P-value |
| Patient age | 1.02 (1.01–1.02) | < 0.001 | 1.01 (1.00 - 1.01) | 0.025 | 1.00 (1.00–1.01) | 0.215 |
| Patient sex (*male* vs *female*) | 0.47 (0.38–0.56) | < 0.001 | 0.42 (0.35–0.50) | < 0.001 | 0.55 (0.46–0.66) | < 0.001 |
| Operator sex (*male* vs *female*) | 0.72 (0.60–0.86) | < 0.001 | 0.72 (0.61–0.86) | < 0.001 | 0.76 (0.65–0.90) | 0.001 |
| Operator experience | 1.07 (0.96–1.20) | 0.215 | 1.00 (0.89–1.11) | 0.934 | 0.93 (0.84–1.03) | 0.176 |

AIC: Akaike Information Criterion.

Both single LA agents and mixtures of different LA led to potentially unsafe doses, with mixtures consistently accounting for the highest proportion of potentially unsafe doses regardless of the calculation method used. This suggests that the practice of mixing LA, while common and intended to combine favorable pharmacological properties such as rapid onset with extended duration, substantially increases the likelihood of exceeding calculated safety thresholds. Furthermore, the calculation rules for determining maximum doses of LA when using mixtures lack standardization [22]. The clinical evidence supporting enhanced efficacy of LA mixtures remains limited [32], and the pharmacokinetic advantages may be offset by the increased complexity of dose calculations [17]. Given these considerations, institutions that routinely use LA mixtures may benefit from implementing standardized protocols or decision-support tools to minimize the risk of inadvertent overdosing.

Demographic factors, including female gender and older age, were associated with a higher likelihood of receiving a potentially unsafe dose. This pattern was consistent across all calculation methods, suggesting that these associations reflect genuine practice patterns rather than artifacts of any specific dosing criteria. It is likely that most physicians did not systematically account for variations in body composition and metabolism when deciding on LA doses [33]. The Akaike Information Criterion comparison revealed that the model using the full conservative calculation rules provided the best fit, followed by IBW and AW, supporting the use of patient-specific factors in dose determination.

The finding that female anesthesiologists were more likely to administer calculated potentially unsafe doses was unexpected and contrasts with established literature showing that female physicians are generally more risk-averse in procedural settings [34,35]. This counterintuitive finding requires careful interpretation and may reflect unmeasured confounders not captured in our retrospective analysis, such as differences in case allocation, patient mix, teaching responsibilities, or other workflow-related factors. Alternatively, it could suggest that gender-based differences in clinical practice may not uniformly apply across all domains of medical decision-making or may be influenced by institutional training patterns and protocols. The retrospective nature of our study prevented us from establishing causality, and this finding highlights the need for prospective research to confirm this association and explore potential underlying mechanisms.

The incidence of LAST was consistent with contemporary literature [4,5]. Vascular punctures, although rare, did not correlate with LAST symptoms. While potentially unsafe doses were frequent, clinical manifestations of toxicity remained rare and within expected ranges. Furthermore, despite the substantial number of potentially unsafe doses, no significant association was observed between these doses and LAST symptoms. This finding is consistent across all four calculation methods and suggests that factors beyond absolute dose play important roles in LAST development. Other genetic and environmental factors likely contribute to the presence or absence of LAST, yet it is essential to recognize that LAST symptoms can easily go unnoticed by healthcare professionals [1]. Moreover, the concomitant use of general anesthesia or deep sedation can mask such symptoms [36]. Additionally, underreporting of such complications during handovers and lack of accurate documentation could represent further contributing factors.

The substantial variation in rates of potentially unsafe doses across the four methods (29.5% to 64.8%) illustrates how different safety criteria can dramatically alter the perception of dosing practices. This variation may explain some of the inconsistencies reported in the literature regarding LA dosing safety. The progressive increase from package insert recommendations to conservative consensus-based rules demonstrates the impact of incorporating patient-specific factors and conservative safety margins. Importantly, none of these calculation methods can be considered a definitive "gold standard", and the clinical relevance of each approach requires further validation through prospective outcome studies.

The consistent pattern of higher rates of potentially unsafe doses with LA mixtures, observed across all calculation methods, highlights opportunities for practice standardization through improved dosing protocols, decision support tools, and targeted educational initiatives. Such interventions could help bridge the gap between pharmacological knowledge and clinical practice while maintaining effective regional anesthesia and reducing practice variability. Rather than suggesting immediate changes to current practices, which appear clinically safe based on the low LAST incidence, these data support the development of more standardized approaches to dose calculation and mixture use.

These findings indicate opportunities for further standardization through enhanced education on LA pharmacokinetics, improved dosing calculators that incorporate a broader range of patient factors, and increased awareness of the complexity associated with mixture calculations [10].

This study has several important limitations that must be considered when interpreting the results. The primary limitation relates to the absence of universally accepted rules for computing maximum safe LA doses. The four calculation methods used, while comprehensive, represent different philosophical approaches to dose determination rather than validated safety thresholds. The conservative consensus-based rules, in particular, may be overly restrictive and could overestimate the frequency of potentially problematic dosing. Our use of Devine's formula for ideal body weight calculation, while standard in our institution, represents another limitation as different weight calculation methods might yield different results. The single-center design limits the generalizability of findings, as dosing practices and patient populations may vary across different institutions and healthcare systems. The retrospective design increases the risk of bias, and selection bias cannot be ruled out since many files were excluded due to incomplete data or use of non-standard LA concentrations. Finally, given the limited incidence of LAST symptoms, no definitive conclusion could be drawn regarding an association between the use of potentially unsafe LA doses and the occurrence of LAST.

Despite these limitations, this study provides valuable insights into the variability of current dosing practices, with rates of potentially unsafe doses varying dramatically depending on the criteria applied. The fact that this occurred in axillary brachial plexus blocks, a commonly performed procedure, suggests similar patterns might exist in other types of regional blocks and at other institutions. However, the absence of clinical correlation between potentially unsafe doses and LAST symptoms across all methods suggests that multiple factors beyond calculated dose thresholds influence clinical outcomes, highlighting the complexity of LAST risk assessment. These findings warrant broader investigation across different block types and practice settings, with particular attention to developing evidence-based dosing protocols and creating standardized approaches to mixture calculations.

## Conclusion

While our study was designed to detect dose differences rather than clinical outcomes, our data reveal considerable variation in current dosing practices that may reflect the complexity of dose determination rather than inadequate safety practices. The most consistent finding across all calculation methods was the higher rates of potentially unsafe doses with LA mixtures compared to single agents. Given the limited evidence for enhanced efficacy of mixtures and the consistent pattern of potentially unsafe doses, this practice deserves careful evaluation and potential standardization. Future research should focus on developing evidence-based dosing protocols, creating standardized approaches to mixture calculations, and conducting multi-center studies to better understand optimal dosing practices in regional anesthesia, with particular emphasis on correlating different calculation methods with clinical outcomes.

## Supporting information

**S1 Checklist. STROBE checklist for the reporting of this observational study.**
(PDF)

## Acknowledgments

The authors have no acknowledgments to declare.

## Author contributions

**Conceptualization:** Mélanie Suppan.

**Data curation:** Mélanie Suppan.

**Formal analysis:** Mélanie Suppan.

**Methodology:** Mélanie Suppan.

**Project administration:** Mélanie Suppan.

**Supervision:** Caroline Flora Samer, Georges Louis Savoldelli.

**Validation:** Mélanie Suppan, Caroline Flora Samer, Georges Louis Savoldelli.

**Visualization:** Mélanie Suppan.

**Writing – original draft:** Mélanie Suppan.

**Writing – review & editing:** Mélanie Suppan, Caroline Flora Samer, Georges Louis Savoldelli.

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
