## [Decision Letter · Decision Letter 0]

14 Jan 2026

Potentially unsafe doses of local anesthetics in axillary brachial plexus block: a single-center retrospective cohort study

PLOS One

Dear Dr. Suppan,

Thank you for submitting your manuscript to PLOS ONE. After careful consideration, we feel that it has merit but does not fully meet PLOS ONE’s publication criteria as it currently stands. Therefore, we invite you to submit a revised version of the manuscript that addresses the points raised during the review process.

**ACADEMIC EDITOR:**
**The reviewers rise important points especially those regarding the methods of the models. The responses would improve the quality of the manuscript, if properly addressed.**

We look forward to receiving your revised manuscript.

Kind regards,

Andrea Cortegiani, M.D.

Academic Editor

PLOS One

Journal Requirements:

[I have read the journal's policy and the authors of this manuscript have the following competing interests: all authors were involved in the development of the LoAD Calc application, which was developed for research purposes only. This application is not monetized and will not be monetized in the future. The calculation rules from this application were used to define the most conservative calculations described in this manuscript.].

4. Please note that your Data Availability Statement is currently missing the repository name. If your manuscript is accepted for publication, you will be asked to provide these details on a very short timeline. We therefore suggest that you provide this information now, though we will not hold up the peer review process if you are unable.

Reviewers' comments:

Reviewer's Responses to Questions

**Comments to the Author**

1. Is the manuscript technically sound, and do the data support the conclusions?

Reviewer #1: Yes

Reviewer #2: Yes

Reviewer #3: Yes

2. Has the statistical analysis been performed appropriately and rigorously?

Reviewer #1: Yes

Reviewer #2: Yes

Reviewer #3: Yes

3. Have the authors made all data underlying the findings in their manuscript fully available?

Reviewer #1: Yes

Reviewer #2: Yes

Reviewer #3: Yes

4. Is the manuscript presented in an intelligible fashion and written in standard English?

Reviewer #1: Yes

Reviewer #2: Yes

Reviewer #3: Yes

Reviewer #1: The article has been very nicely written with all data and study methodology being done in an appropriate matter. The language is simple yet precisely written. The article covers the area of a widely known subject regarding the use of anesthetics and its potential adverse effects seen throughout.

Reviewer #2: Overall, the manuscript presents a well-conducted retrospective cohort study on potentially unsafe local anesthetic doses in axillary brachial plexus blocks. The use of four progressively conservative dosing methods is a particular strength and adds significant rigor.

I have the following minor comments for improvement:

1. Sample size calculation: The study assumes detection of a 1% dose difference with 0.0022 precision. Clarify how this translates to the primary outcome (proportion of unsafe doses across methods).

2. Table 1: Specify whether continuous variables are reported as mean ± SD or median (IQR).

3. Comorbidities/treatments: In the Table 1 legend, list the most common treatments altering LA metabolism (e.g., CYP inhibitors, heart failure meds).

4. Figure 2: Shows unsafe dose proportions for single agents (most conservative method). Consider adding similar panels for mixtures and other methods, with p-values, to enhance visual comparison.

5. Multivariate logistic regression: Only 4 variables were included. Justify exclusion of others (e.g., weight, comorbidities, interacting drugs)—were they tested in univariate analysis or omitted for clinical/logical reasons?

6. LAST symptom identification: Elaborate on how LAST symptoms were extracted and classified from records (e.g., keyword search, clinical criteria, inter-rater agreement).

7. Power for LAST association: With only 19 LAST events (0.79%), the analysis may be underpowered. Acknowledge this limitation or provide a post-hoc power calculation.

These minor clarifications will strengthen an already robust and clinically relevant manuscript

Reviewer #3: Thank you for the opportunity to review this manuscript. I found the topic highly relevant, especially because safe” local anesthetic dosing is a challenge in everyday decision making. The authors have done an impressive job assembling a large dataset and applying multiple dosing frameworks. The manuscript is clear and well-organized. Still, there are several areas where more explanation would help readers fully understand the reasoning behind the methods and the implications of the findings.

1. One of the main strengths of this study is that it compares LA doses using four different calculation methods, from the simplest (package-insert values) to the most conservative individualized approach. However, the manuscript does not fully explain why these four specific criteria were selected. A brief explanation of how each method reflects real-world practice, why they differ, and what the authors hoped to learn from comparing them would really help orient readers. Even two or three sentences explaining the clinical relevance, why these methods matter to anesthesiologists making dosing decisions, would make this section much more accessible.

2. I also think the manuscript would benefit from a short discussion about the rationale for mixing LAs. This is a common practice in some institutions but not in others, and there is still debate about whether mixing offers meaningful clinical advantages. In the methods, the authors apply the principle that systemic toxicity is additive, which is reasonable, but readers may wonder whether mixing agents should even be encouraged or if it introduces unnecessary complexity. Explaining the pharmacological reasoning and briefly mentioning what the literature says about the benefits or risks of mixing would be more relevant. This does not need to be long, just enough to show that the authors have considered why mixtures occur and what is known about their safety profile.

3. Statistical analysis is straightforward, but the use of AIC needs a bit more explanation. AIC is perfectly acceptable for comparing models, but readers may not immediately understand why it was chosen here or how it helps interpret the results. A short justification makes this clearer. Similarly, since the study looks at factors like patient age, patient sex, operator experience, and operator sex, it would be helpful to say whether the authors explored any potential interactions. In a setting like this, certain variables may influence each other, and even a brief statement acknowledging whether interactions were tested or why they were not would improve confidence in the model.

Overall, this manuscript tackles an important and practical question, and the dataset is strong. Addressing the points above would make the work even clearer and more useful for clinicians who rely on these dosing principles every day.

**Do you want your identity to be public for this peer review?** For information about this choice, including consent withdrawal, please see our Privacy Policy

Reviewer #1: No

Reviewer #2: **Yes:** Rohit Agrawal

Reviewer #3: No

---

## [Author Response · Author response to Decision Letter 1]

27 Jan 2026

Editorial comments:

Comment 1. Please ensure that your manuscript meets PLOS ONE's style requirements, including those for file naming. The PLOS ONE style templates can be found at

Response: We have carefully reviewed PLOS ONE's style requirements and believe our manuscript complies with the journal's formatting guidelines, including file naming conventions. Should any specific elements require adjustment, we would be happy to make the necessary changes.

Comment 2. Please provide additional details regarding participant consent. In the ethics statement in the Methods and online submission information, please ensure that you have specified what type you obtained (for instance, written or verbal, and if verbal, how it was documented and witnessed). If your study included minors, state whether you obtained consent from parents or guardians. If the need for consent was waived by the ethics committee, please include this information.

Response: We thank you for this request and apologize for the lack of clarity regarding participant consent. In our original manuscript, the information regarding consent was dispersed across two sections of the Methods (Study design and Inclusion and exclusion criteria), which may have caused confusion.

To clarify:

• Our study only included adult patients (aged 18 years or older); no minors were included.

• A written general consent form for data reuse has been in place at our institution (Geneva University Hospitals) since 2017.

• However, given the retrospective nature of this study using anonymized medical records that posed no potential harm to participants, the regional ethics committee (Commission Cantonale d'Ethique de la Recherche, Geneva, Switzerland) waived the requirement for individual informed consent and authorized data reuse even for patients who had not signed the general consent form.

We have now consolidated and clarified this information in the Study design section of the Methods as follows: “Approval by the regional ethics committee (Commission Cantonale d'Ethique de la Recherche CCER - Req 2022-01195, Geneva, Switzerland, Chairperson Prof B. Hirschel) was obtained on 16/08/2022. Only adult patients (aged 18 or older) were included in this study. A written general consent form for data reuse has been in place at our institution since 2017. However, given the retrospective nature of this study using anonymized medical records that posed no potential harm to participants, the ethics committee waived the requirement for individual informed consent and authorized data reuse even without a signed general consent form.”

The corresponding information has been removed from the Inclusion and exclusion criteria section to avoid redundancy.

Comment 3. Thank you for stating the following in the Competing Interests section:

[I have read the journal's policy and the authors of this manuscript have the following competing interests: all authors were involved in the development of the LoAD Calc application, which was developed for research purposes only. This application is not monetized and will not be monetized in the future. The calculation rules from this application were used to define the most conservative calculations described in this manuscript.].

Response: We confirm that our competing interests do not alter our adherence to all PLOS ONE policies on sharing data and materials. There are no restrictions on sharing of data or materials. We have updated our Competing Interests statement as follows: “All authors were involved in the development of the LoAD Calc application, which was developed for research purposes only. This application is not monetized and will not be monetized in the future. The calculation rules from this application were used to define the most conservative calculations described in this manuscript. This does not alter our adherence to PLOS ONE policies on sharing data and materials.”

This updated statement has been included in our cover letter.

Comment 4. Please note that your Data Availability Statement is currently missing the repository name. If your manuscript is accepted for publication, you will be asked to provide these details on a very short timeline. We therefore suggest that you provide this information now, though we will not hold up the peer review process if you are unable.

Response: We thank you for pointing this out. Our anonymized dataset has been deposited in the Yareta repository, which is the institutional research data repository of the University of Geneva. The complete Data Availability Statement is as follows: “The anonymized dataset containing all data of the patients included in the analysis is available in the Yareta repository (University of Geneva) at: https://doi.org/10.26037/YARETA:3HCQXRX5ZFGULNWRCIBUCJGZIU”

Reviewers' comments:

Reviewer #1:

Comment 1: The article has been very nicely written with all data and study methodology being done in an appropriate matter. The language is simple yet precisely written. The article covers the area of a widely known subject regarding the use of anesthetics and its potential adverse effects seen throughout.

Response: We are delighted by your positive feedback and sincerely thank you for your kind words regarding our manuscript. It is very rewarding to know that you found our methodology appropriate and our writing clear and accessible. We truly appreciate the time and effort you dedicated to reviewing our work.

Reviewer #2:

Comment 1: Overall, the manuscript presents a well-conducted retrospective cohort study on potentially unsafe local anesthetic doses in axillary brachial plexus blocks. The use of four progressively conservative dosing methods is a particular strength and adds significant rigor.

I have the following minor comments for improvement:

Response: We sincerely thank you for your positive evaluation of our manuscript and for recognizing the rigor of our methodology, particularly the use of four progressively conservative dosing methods. We appreciate the time and effort you invested in providing constructive feedback. Below, we address each of the points you raised and detail the corresponding revisions we have made to strengthen the manuscript.

Comment 2: Sample size calculation: The study assumes detection of a 1% dose difference with 0.0022 precision. Clarify how this translates to the primary outcome (proportion of unsafe doses across methods).

Response: Thank you for this comment. Indeed, power analyses are rather uncommon in retrospective studies, and our aim was to ensure that we would be able to include enough patients to detect significant differences provided they existed. We therefore used the formula for sample size calculation in prevalence studies as described by Pourhoseingholi et al. (PMID: 24834239, reference n° 20). The formula proposed in this article is: n = Z2P(1-P)/d2, where n is the sample size, Z the confidence level, P the expected prevalence and d the precision – in this case, we used P/5 to compute it). Thus, the calculation was performed to detect a 1% difference with a precision of 0.002 and 95% power (n = 0.952*0.01*(1-0.01)/0.0022 = 2233.6875, rounded to 2234), which makes the results particularly robust. To clarify this aspect, the following sentence was added to the methods section: “To avoid fragility, precision was increased by dividing the estimated prevalence (0.01) by 5.”

Comment 3: Table 1: Specify whether continuous variables are reported as mean ± SD or median (IQR).

Response: We thank you for this observation. Continuous variables are reported as mean ± SD, as stated in the Statistical Analysis section. We have added “±SD” where appropriate in Table 1. Furthermore, your comment made us realize that the “n, %” legends were sometimes misplaced or even missing, and therefore enabled us to correct this oversight.

Comment 4: Comorbidities/treatments: In the Table 1 legend, list the most common treatments altering LA metabolism (e.g., CYP inhibitors, heart failure meds).

Response: The following point was added to the legend: “5Major CYP1A2 inhibitors such as ciprofloxacin, norfloxacin, and fluvoxamine, and major CYP3A inhibitors, such as azole antifungals, macrolides, calcium channel blockers, HIV antiretroviral therapy, and tyrosine kinase inhibitors.”

Comment 5: Figure 2: Shows unsafe dose proportions for single agents (most conservative method). Consider adding similar panels for mixtures and other methods, with p-values, to enhance visual comparison.

Response: Thank you for this suggestion. We carefully considered adding similar panels for mixtures and other calculation methods. However, we decided against this for two reasons. First, the vast majority of mixtures (97.6%) consisted of ropivacaine with lidocaine, which would limit meaningful visual comparison across mixture types. Second, we were concerned that adding multiple panels might overwhelm the reader and detract from the main message. The detailed breakdown of potentially unsafe doses for mixtures and across all calculation methods is provided in the text and in Table 2. Should you or the Editor feel that additional figures would enhance the manuscript, we would be happy to reconsider.

Comment 6: Multivariate logistic regression: Only 4 variables were included. Justify exclusion of others (e.g., weight, comorbidities, interacting drugs)—were they tested in univariate analysis or omitted for clinical/logical reasons?

Response: You are right, this is an important question. The four variables included (patient age, patient sex, operator sex, operator experience) were selected a priori based on clinical relevance and data availability. Weight was not included as an independent predictor because it is already incorporated into the outcome calculation itself (potentially unsafe dose is weight-dependent), which would introduce collinearity. Similarly, comorbidities and interacting drugs are embedded in the most conservative calculation method and were therefore not analyzed as separate predictors for the same reason. We have clarified this in the Methods section: “The variables were selected according to their clinical relevance and to data availability. Weight, comorbidities, and interacting drugs were not included as independent predictors because they are already incorporated into the outcome calculation, which would introduce collinearity.”

Comment 7: LAST symptom identification: Elaborate on how LAST symptoms were extracted and classified from records (e.g., keyword search, clinical criteria, inter-rater agreement).

Response: We thank you for this comment. Most of this information is already provided in the Methods section. All files were manually reviewed to search for LAST symptoms, as stated: “This included a review of intraoperative and follow-up notes to identify the presence of potential LAST symptoms.” Classification was carried out according to established criteria in the literature, with symptoms categorized as “mild-to-moderate (perioral numbness, metallic taste, confusion, muscle twitching, etc.) or severe (seizures, loss of consciousness, respiratory depression, cardiac arrhythmias, severe hypotension, and cardiac arrest).” To further clarify, we have slightly revised the wording to specify that files were “manually reviewed”.

Comment 8: Power for LAST association: With only 19 LAST events (0.79%), the analysis may be underpowered. Acknowledge this limitation or provide a post-hoc power calculation.

Response: We absolutely agree with you on this account. The following limitation has been added as advised: “Finally, given the limited incidence of LAST symptoms, no definitive conclusion could be drawn regarding an association between the use of potentially unsafe LA doses and the occurrence of LAST.”

Comment 9: These minor clarifications will strengthen an already robust and clinically relevant manuscript

Response: We thank you once again for your time and comments and believe that our manuscript has indeed been enhanced by virtue of the review process.

Reviewer #3:

Comment 1: Thank you for the opportunity to review this manuscript. I found the topic highly relevant, especially because safe local anesthetic dosing is a challenge in everyday decision making. The authors have done an impressive job assembling a large dataset and applying multiple dosing frameworks. The manuscript is clear and well-organized. Still, there are several areas where more explanation would help readers fully understand the reasoning behind the methods and the implications of the findings.

Response: We sincerely thank you for your thoughtful evaluation and for recognizing the clinical relevance of our work. We are grateful for the time you dedicated to providing detailed and constructive feedback. Below, we address each of your comments and describe the revisions we have made to improve the manuscript.

Comment 2: One of the main strengths of this study is that it compares LA doses using four different calculation methods, from the simplest (package-insert values) to the most conservative individualized approach. However, the manuscript does not fully explain why these four specific criteria were selected. A brief explanation of how each method reflects real-world practice, why they differ, and what the authors hoped to learn from comparing them would really help orient readers. Even two or three sentences explaining the clinical relevance, why these methods matter to anesthesiologists making dosing decisions, would make this section much more accessible.

Response: We thank you for this valuable suggestion. We agree that explaining the rationale behind our choice of four calculation methods would help orient readers. We have added the following paragraph to the Methods section:

“Four different calculation methods were selected to represent a spectrum of approaches encountered in clinical practice, from the simplest to the most individualized. These range from basic package insert recommendations, commonly used as a default reference, to actual weight-based calculations reflecting routine clinical practice, to ideal body weight-based calculations addressing pharmacokinetic concerns in overweight and obese patients, and finally to conservative consensus-based rules incorporating patient-specific factors known to affect LA metabolism (age, organ dysfunction, drug interactions) that are rarely systematically considered in everyday practice. Comparing these progressively more conservative approaches allows clinicians to benchmark their own institutional practices against multiple standards and illustrates how the assessment of dosing safety varies depending on the criteria applied.”

Comment 3: I also think the manuscript would benefit from a short discussion about the rationale for mixing LAs. This is a common practice in some institutions but not in others, and there is still debate about whether mixing offers meaningful clinical advantages. In the methods, the authors apply the principle that systemic toxicity is additive, which is reasonable, but readers may wonder whether mixing agents should even be encouraged or if it introduces unnecessary complexity. Explaining the pharmacological reasoning and briefly mentioning what the literature says about the benefits or risks of mixing would be more relevant. This does not need to be long, just enough to show that the authors

---

## [Decision Letter · Decision Letter 1]

16 Feb 2026

Dear Dr. Suppan,

Thank you for submitting your manuscript to PLOS ONE. After careful consideration, we feel that it has merit but does not fully meet PLOS ONE’s publication criteria as it currently stands. Therefore, we invite you to submit a revised version of the manuscript that addresses the points raised during the review process.

We look forward to receiving your revised manuscript.

Kind regards,

Andrea Cortegiani, M.D.

Academic Editor

PLOS One

Journal Requirements:

Reviewers' comments:

Reviewer's Responses to Questions

**Comments to the Author**

Reviewer #3: (No Response)

2. Is the manuscript technically sound, and do the data support the conclusions?

Reviewer #3: Yes

3. Has the statistical analysis been performed appropriately and rigorously?

Reviewer #3: Yes

4. Have the authors made all data underlying the findings in their manuscript fully available?

Reviewer #3: Yes

5. Is the manuscript presented in an intelligible fashion and written in standard English?

Reviewer #3: Yes

Reviewer #3: This is a well designed retrospective Cohort study that addresses an important and clinically relevant question. The manuscript is well written and structured, however, if few points are interpreted with clarity this strengthens the manuscript.

The finding that female patients and female anesthesiologists were more likely to be associated with potentially unsafe dosing is interesting but sensitive. Although the Discussion acknowledges possible unmeasured confounding (lines 347-355), this finding is presented prominently in the Abstract without sufficient qualification. The language in the Abstract should be softened to clearly state that these associations are observational and unexplained. In the Discussion, consider briefly expanding on potential structural or workflow-related explanations (e.g., differences in case mix, teaching roles, or patient allocation) to avoid overly individual-level interpretations.

Also, the manuscript uses multiple terms such as “potentially unsafe dose,” “dose exceeding recommendations,” and “overdose.” For clarity, consider defining one preferred term early and using it consistently throughout the manuscript.

**Do you want your identity to be public for this peer review?** For information about this choice, including consent withdrawal, please see our Privacy Policy

Reviewer #3: **Yes:** Kumud Chapagain

---

## [Author Response · Author response to Decision Letter 2]

22 Feb 2026

Dear Editor,

We are grateful for the opportunity to revise our manuscript entitled “Potentially unsafe doses of local anesthetics in axillary brachial plexus block: a single-center retrospective cohort study” (Manuscript ID: PONE-D-25-55020R1) and would like to sincerely thank you and the reviewers for their time and thoughtful evaluation of our work. The comments provided have been valuable in helping us improve the clarity and overall quality of the manuscript. We have carefully considered each point raised and have revised the manuscript accordingly. We believe the revised version is significantly strengthened and hope it now meets the standards required for publication in Plos One.

A point-by-point response to each comment is provided below. All changes made to the manuscript are highlighted in the revised version.

Reviewers' comments:

Reviewer #3:

Comment 1: This is a well designed retrospective Cohort study that addresses an important and clinically relevant question. The manuscript is well written and structured, however, if few points are interpreted with clarity this strengthens the manuscript.

Response: Thank you for this positive assessment of our work. We have carefully addressed the specific points raised in the subsequent comments and believe these revisions strengthen the manuscript.

Comment 2: The finding that female patients and female anesthesiologists were more likely to be associated with potentially unsafe dosing is interesting but sensitive. Although the Discussion acknowledges possible unmeasured confounding (lines 347-355), this finding is presented prominently in the Abstract without sufficient qualification. The language in the Abstract should be softened to clearly state that these associations are observational and unexplained. In the Discussion, consider briefly expanding on potential structural or workflow-related explanations (e.g., differences in case mix, teaching roles, or patient allocation) to avoid overly individual-level interpretations.

Response: Thank you for raising this point. We agree that these associations are difficult to adequately qualify within the constraints of an abstract and have therefore removed this finding from the Abstract, where it could be misinterpreted. It remains presented with appropriate qualification in the Discussion, which we have expanded to include potential structural and workflow-level explanations as suggested.

The revised paragraph now reads: “The finding that female anesthesiologists were more likely to administer potentially unsafe doses was unexpected and contrasts with established literature showing that female physicians are generally more risk-averse in procedural settings [34,35]. This counterintuitive finding requires careful interpretation and may reflect unmeasured confounders not captured in our retrospective analysis, such as differences in case allocation, patient mix, teaching responsibilities, or other workflow-related factors. Alternatively, it could suggest that gender-based differences in clinical practice may not uniformly apply across all domains of medical decision-making or may be influenced by institutional training patterns and protocols. The retrospective nature of our study prevented us from establishing causality, and this finding highlights the need for prospective research to confirm this association and explore potential underlying mechanisms.”

Comment 3: Also, the manuscript uses multiple terms such as “potentially unsafe dose,” “dose exceeding recommendations,” and “overdose.” For clarity, consider defining one preferred term early and using it consistently throughout the manuscript.

Response: Thank you for this observation. We have adopted “potentially unsafe dose” as the preferred term throughout the manuscript, including in the Methods section, as it best reflects the nuanced, calculation-based nature of our findings without implying that clinical toxicity will necessarily occur. All instances of alternative terminology have been replaced accordingly.

---

## [Editor Report · Decision Letter 2]

25 Feb 2026

Potentially unsafe doses of local anesthetics in axillary brachial plexus block: a single-center retrospective cohort study

PONE-D-25-55020R2

Dear Dr. Suppan,

We’re pleased to inform you that your manuscript has been judged scientifically suitable for publication and will be formally accepted for publication once it meets all outstanding technical requirements.

Kind regards,

Andrea Cortegiani, M.D.

Academic Editor

PLOS One
---

## [Editor Report · Acceptance letter]

PONE-D-25-55020R2

PLOS One

Dear Dr. Suppan,

I'm pleased to inform you that your manuscript has been deemed suitable for publication in PLOS One. Congratulations! Your manuscript is now being handed over to our production team.

Kind regards,

on behalf of

Dr. Andrea Cortegiani

Academic Editor

PLOS One